# Pan-cancer analysis of the effect of biopsy site on tumor mutational burden observations

Simon Papillon-Cavanagh [1✉], Julia F. Hopkins[2], Shakti H. Ramkissoon[2], Lee A. Albacker [2] & Alice M. Walsh[1]

## Abstract

**Background** Tumor mutational burden (TMB) has been proposed as a predictive biomarker of response to immunotherapy. Efforts to standardize TMB scores for use in the clinic and to identify the factors that could impact TMB scores are of high importance. However, the biopsy collection site has not been assessed as a factor that may influence TMB scores.

**Methods** We examine a real-world cohort comprising 137,771 specimens across 47 tissues in 12 indications profiled by the FoundationOne assay (Foundation Medicine, Cambridge, MA) to assess the prevalence of biopsy sites for each indication and their TMB scores distribution.

**Results** We observe a wide variety of biopsy sites from which specimens are sent for genomic testing and show that TMB scores differ in a cancer- and tissue-specific manner. For example, brain or adrenal gland specimens from NSCLC patients show higher TMB scores than local lung specimens (mean difference 3.31 mut/Mb; $p < 0.01$, 3.90 mut/Mb; $p < 0.01$, respectively), whereas bone specimens show no difference.

**Conclusions** Our data shed light on the biopsied tissue as a driver of TMB measurement variability in clinical practice.

## Plain language summary

The total number of mutations, or changes, found within the DNA of cancer cells in a tumor sample is known as the tumor mutational burden (TMB). TMB scores have been proposed to be a marker of how well a tumor might respond to immunotherapy, a type of treatment that triggers the body's immune system to target the cancer. Here, we look at whether TMB scores are dependent on the location in the body from which a tumor sample is taken, the biopsy site. We use over 100,000 specimens from 47 tissues for 12 different cancer types and test for associations between TMB scores and biopsy site. We find that the biopsy site has a strong impact on TMB score. These findings might impact on how clinicians interpret the results of genetic testing in patients and how they make decisions on treatment.

[1] Informatics and Predictive Sciences, Bristol-Myers Squibb Co, Princeton, NJ, USA. [2] Foundation Medicine, Inc., Cambridge, MA, USA.
✉email: simon.papillon-cavanagh@bms.com

mmunomodulatory cancer drugs, such as anti-programmed death-1 (anti-PD-1) antibodies, have transformed the clinical oncology landscape, leading to significant clinical benefit across multiple cancer types. However, not all patients benefit, highlighting the need for predictive biomarkers to guide clinical decision-making. Tumor mutational burden (TMB), a proxy for tumor-specific neoantigens leading to recognition by cytotoxic T cells, has been proposed to stratify patients likely to respond to anti-PD-1 therapy. TMB score is defined as the number of somatic nonsynonymous mutations per megabase (mut/Mb), as assessed by next-generation sequencing of targeted genomic regions.

TMB score has been reported to be associated with known mutagenic processes such as deficient DNA mismatch repair, smoking, and ultraviolet light exposure[1] and in some cases, chemotherapeutic treatment[2]. Within a cancer type, TMB scores can vary widely, with melanoma patients showing scores ranging from 1 to 1000 mut/Mb[3]. Previous work has shown that there were differences in TMB between primary and metastatic tissues, with metastatic tumors having higher TMB scores[4].

High TMB has previously been associated with anti-PD-1 response in multiple cancer types, most notably in non-small cell lung cancer (NSCLC)[5–7] and melanoma[8], indications in which TMB values are higher than most cancers[3]. Those associations with long-term survival benefits have led to attempts at identifying TMB-high patients in multiple indications accompanied by efforts to identify cancer-specific thresholds[9]. Recently, TMB was shown to correlate with higher response rates in patients treated with anti-PD-1 across multiple cancer types, leading to an FDA approval for tissue TMB-high patients with solid tumors. This approval was based on a threshold of 10 mut/Mb, applicable to all solid cancer types[10].

In parallel, important efforts have been made to develop TMB as a standardized and cost-effective clinical assay leading to a better understanding of how to best quantify and interpret TMB as a biomarker[11]. These efforts, however, have been focused on the technical aspects of TMB measurements, such as panel size, sequencing depth, bioinformatics pipelines and variant filtering and have ignored the source of the specimen as a potential factor influencing TMB.

Profiling the genetic diversity in cancer patients has been an active field of study across multiple cancer types. Large consortium efforts, such as the TRACERx initiative, have shown that within a single tumor lesion, there is considerable genetic heterogeneity, which can affect patient outcomes[12,13]. Focused on primary versus metastatic diversity, a study in NSCLC patients showed that *EGFR* expression was lower in metastatic tissues[14]. In contrast, primary and metastatic specimens from breast cancer patients had a high degree of concordance between the immunohistochemistry staining levels of estrogen receptor, progesterone receptor, HER-2, and Ki-67[15,16]. More recent work identified differences in TMB measurements between specimens from primary and metastatic biopsies[4]. Moreover, in a recent study profiling TMB measurements in specimens from lung adenocarcinoma patients, the authors showed that in addition to metastasis-specific differences, there were site-specific differences, notably in brain and adrenal gland metastases[17]. Here we extend those previous efforts and profile the effect of the biopsy site on TMB measurements in 137,771 specimens, biopsied from 47 tissues across 12 cancer types in a real-world cohort with genomic sequencing of tissue from Foundation Medicine (Cambridge, MA). Our comprehensive study shows that the biopsy site is associated with TMB score, with cancer-specific patterns.

## Methods

**Cohort and sample selection**. We selected patient specimens profiled on bait sets DX1, D2, T5a, or T7 of the Foundation Medicine FoundationOne CDx or FoundationOne assay, as they are performed on tumor material (as opposed to blood). In cases where a patient had multiple specimens available, we kept the most recently collected specimen. Specimens for the same patients collected on the same date were chosen arbitrarily. To reduce the impact of tumor purity on results, each specimen was hand reviewed by a pathologist to ensure it was suitable for sequencing and we kept only specimens with ≥30% tumor purity, resulting in a collection of high-quality samples with a median coverage exceeding 500×. We filtered mutations with an allele frequency exceeding 5%. To reduce modeling noise and multiple testing, in each cancer type independently, we discarded specimens from tissues that had <50 specimens in total. Biopsy sites were assigned as primary by manual curation based on prior knowledge of each tumor type. Ethical approval, including a waiver of informed consent and a HIPAA waiver of authorization, was received from the Western Institutional Review Board (Protocol No. 20152817). Consented data that can be released are included in the article and its supplementary files. Patients were not consented for the public release of underlying sequence data.

**Statistical association with TMB**. We used the TMB score as previously defined[18]. Given that TMB score is log-normal distributed, we log-transformed the TMB score variable ($\log(\text{TMB} + 1)$). Each cancer type was treated independently. We used multivariate linear regression and estimated marginal means to compute the difference in TMB score associated with the biopsied site. We reported the difference between all tissues and a reference/primary site, which differed by cancer type.

Difference estimates and confidence intervals (CIs) were back-transformed on "TMB scale (mut/Mb)" using re-gridding provided in the R package *emmeans*[19,20]. In order to control for sample quality factors that may influence the TMB score, we included covariates that measured the median sample sequencing coverage, tumor purity as estimated by a pathologist, and tumor purity based on FoundationMedicine computational model[21].

**Percentage of patients assessed as TMB-high**. We grouped specimens into TMB-high and TMB-low categories based on a 10 mut/Mb cutoff and calculated the difference in percentage and odds ratio (OR) for all tissues compared to a reference/primary site, which differed by cancer type. We tested for association using chi-square test. If a cancer type–tissue pair had <10 TMB-high or TMB-low specimens, the comparison was not performed.

**Controlling for the presence of metastasis**. We used available clinical annotations from the Flatiron-Foundation Medicine Clinico-Genomic Database (CGDB; Flatiron Health, New York, NY) from breast, non-small cell lung, and melanoma cancers to stratify patients according to whether they had a metastasis reported prior to primary specimen collection. Each cancer type and target specimen type were modeled independently, as described above. Sample counts for each comparison can be found in Supplementary Data 5.

**Tissue pair analysis**. We aggregated all specimens by patients and filtered for patients who had at least 2 biopsies from different sites in a 90 days window. The most frequent tissue pair was primary lung and metastatic brain specimens in lung cancer patients, with nine patients.

**Reporting summary**. Further information on research design is available in the Nature Research Reporting Summary linked to this article.

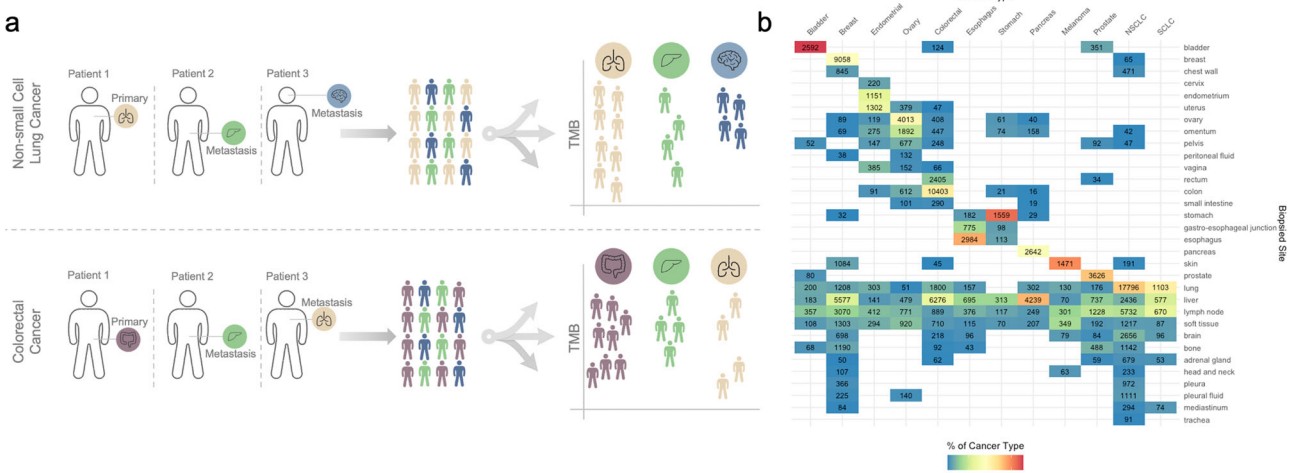

**Fig. 1 Overview of approach and biopsied sites. a** Schematic representation of the approach. For each cancer type independently, patients were separated into groups according to the biopsied site of their specimen used for genomic testing. **b** Distribution of sample counts for each biopsied site and cancer type used for analysis. The color gradient represents the percentage of samples from a specific site within a cancer type.

## Results

**Specimens sent for genomic testing vary**. In a real-world setting, specimens used for genomic screening may come from any tumor lesion. Thus, we first profiled the distribution of specimen sites submitted for comprehensive genomic profiling (Fig. 1a). We observed heterogeneity in biopsied sites within different cancer types, consistent with metastatic patterns[22]. Specimens from bladder and stomach cancers were largely biopsied from their local site (68.9%, 2592/3760; 61.7% 1559/2525, respectively), whereas those from breast cancer were mostly from metastatic sites. In detail, breast cancer specimens originated from the local site (35.5%, 9058/25,492), liver metastases (21.9%, 5577/25,492), or lymph node (12.0%, 3070/25,492) metastases (Fig. 1b, Supplementary Data 1). Other cancers, such as colorectal, endometrial, and esophageal/esophagogastric junction cancers, also had specimens from diverse tissues in accordance with their multiple primary sites.

**Metastatic and primary TMB scores differ in some cancer types**. Next, we compared the TMB scores from metastatic and primary tissues, grouping all metastatic specimens together. We found that TMB scores in metastatic lesions varied by cancer type. Metastatic specimens from SCLC patients had lower TMB scores (mean difference −0.66 mut/Mb; $p < 0.01$; Fig. 2a). In contrast, metastatic specimens from NSCLC patients showed modestly higher TMB scores (mean difference 0.53 mut/Mb; $p < 0.01$; Fig. 2a). In endometrial and breast cancer patients, we observed strikingly higher TMB scores in metastases compared to specimens from primary lesions (mean difference 0.88 mut/Mb, $p < 0.01$; 0.79 mut/Mb, $p < 0.01$, respectively). We observed a nearly significant difference in bladder cancers (mean difference 0.51 mut/Mb, $p < 0.05$) and no differences in patients with stomach or esophageal cancers.

**TMB score is associated with biopsied site**. Next, we profiled the TMB scores in a more granular manner by dividing specimens according to their biopsied sites (Fig. 1a). We observed different TMB score distributions across different sites, such as in specimens from adrenal glands and brain sites in NSCLC patients (Fig. 2b, c). Consequently, the fraction of patients assessed as TMB-high based on a TMB score cutoff of 10 mut/Mb was significantly higher in brain (56.8%; 1508/2656, $p < 0.01$) and adrenal gland specimens (60.5%; 411/679, $p < 0.01$) versus lung

specimens (36.9%; 6375/17,796), suggesting that the biopsied site can potentially influence clinical decision-making.

Using a multivariate linear model to control for potential confounders, such as tumor purity and sequencing depth, we observed similar biopsy site-specific differences in TMB scores relative to a reference primary tissue across multiple cancer types (Fig. 2d). As described above, specimens from NSCLC patients biopsied from the adrenal gland and brain metastases had strikingly higher TMB scores relative to lung biopsies (mean difference 3.90 mut/Mb; 95% CI [2.55–5.24]; $p < 0.01$, 3.31 mut/Mb; 95% CI [2.62–4.00]; $p < 0.01$, respectively), whereas liver or bone metastases were not significantly different (Fig. 2d and Supplementary Data 2). Consistent with NSCLC, brain metastases were also associated with higher TMB scores in breast cancer patients (mean difference 1.82 mut/Mb; 95% CI [1.25–2.40]; $p < 0.01$). In contrast, liver specimens from breast and SCLC showed higher and lower TMB scores (mean difference 0.64 mut/Mb; 95% CI [0.45 to 0.83]; $p < 0.01$, −1.59 mut/Mb; 95% CI [−2.39 to −0.79]; $p < 0.01$, respectively), relative to their respective primary tissue.

Our results showed that the observed differences in TMB scores between local and metastatic specimens are confounded by combining multiple metastatic locations with diverse TMB patterns. For instance, overall, TMB scores from bladder cancer metastatic specimens only marginally differed from primary specimens (Fig. 2a). However, lung metastases had significantly lower TMB scores than bladder specimens (mean difference −1.33 mut/Mb; 95% CI [−2.62 to −0.14]; $p = 0.036$), whereas lymph node specimens show higher TMB scores (mean difference 2.65 mut/Mb; 95% CI [1.14–4.17]; $p < 0.01$). Thus, in aggregate, those two sites cancel each other out.

**Tissue-specific TMB difference impact TMB-high status**. Next, we sought to quantify the impact of the tissue-specific TMB patterns on the proportion of specimens that would be assessed as TMB-high, using a cutoff of 10 mut/Mb, across all cancer types. Globally, the impact of the biopsy site on the proportion of TMB-high samples was highly consistent with our previous analysis using TMB as a continuous variable with the largest difference in percentage corresponding the largest difference in average mut/Mb (Fig. 2d, e). Surprisingly, we found that even in cancer types with overall low TMB, the likelihood of observing a TMB-high sample was meaningfully impacted by the site of biopsy (Fig. 2e

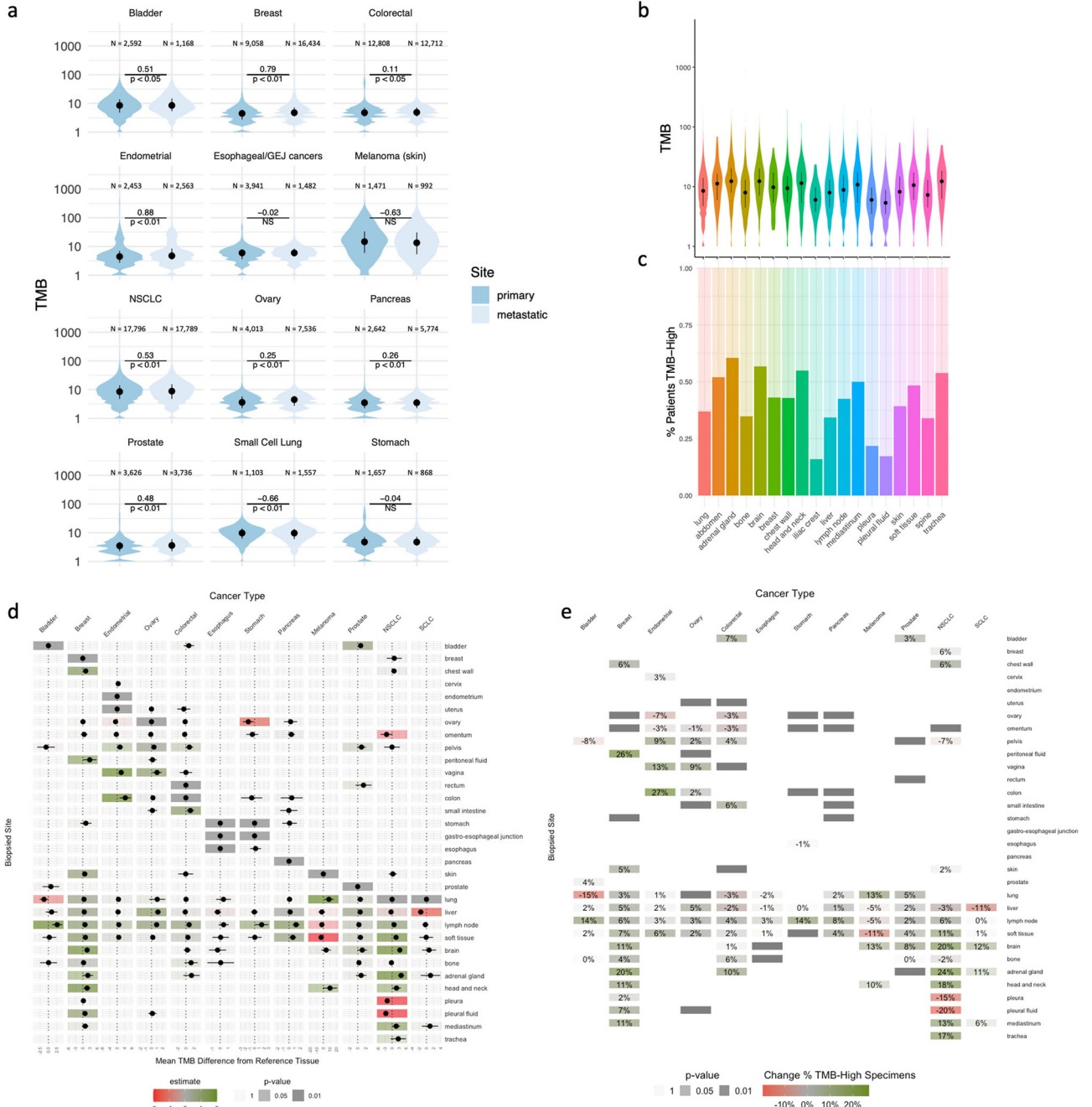

**Fig. 2 TMB score varies by biopsied site. a** Comparison of TMB score distributions in metastases and primary tissues. The average difference and significance computed from a multivariate linear model are shown. Patient counts for each cancer type and biopsy site are provided in Fig. 1b and Supplementary Data 1. **b** Distribution of TMB scores across biopsied sites in patients diagnosed with NSCLC. Patient counts for each cancer type and biopsy site are provided in Fig. 1b and Supplementary Data 1. **c** Distribution of the fraction of NSCLC patients assessed as TMB-high as a function of the biopsied sites used for genomic testing. TMB-high cutoff ≥10 mut/Mb. Patient counts for each cancer type and biopsy site are provided in Fig. 1b and Supplementary Data 1. **d** Forest plots from multivariate linear models testing the association between the biopsied site (rows) and TMB scores across different cancer types (columns). Each cancer type is modeled independently. Reference tissues used for modelling are represented by a dark gray box with a dot at 0. The dots and color gradient represent the average difference between the tissue and a reference tissue. The horizontal lines represent the 95% confidence interval of the average difference. The transparency of the box colors is adjusted according to the significance (p value) of the association. Patient counts for each cancer type and biopsy site are provided in Fig. 1b and Supplementary Data 1. **e** Heatmap showing differences of percentage of specimens assessed as TMB-high (≥10 mut/Mb) between the biopsied site (rows) and reference tissue, across different cancer types (columns). The transparency of the box colors is adjusted according to the significance (p value) of the difference assessed by chi-square test. Patient counts for each cancer type and biopsy site are provided in Fig. 1b and Supplementary Data 1.

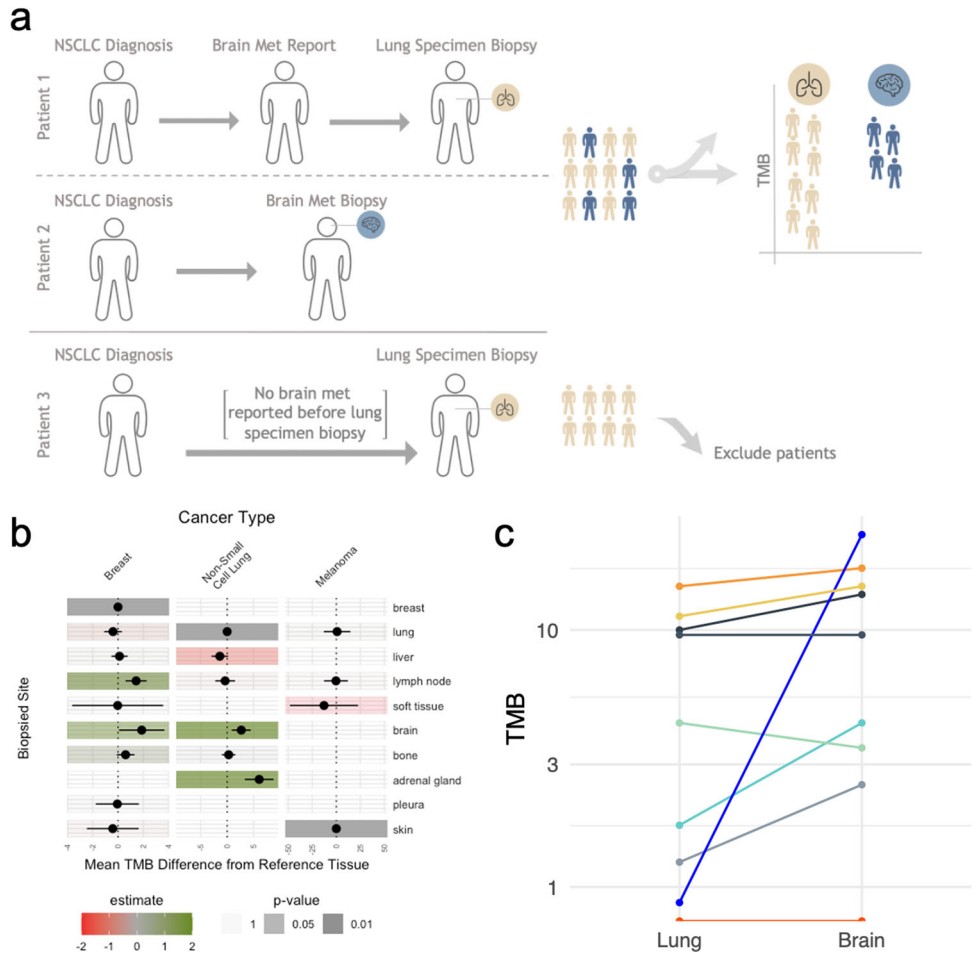

**Fig. 3 Differences in TMB between metastatic and primary tumors. a** Schematic representation of patient stratification. For each metastatic site, we kept patients with primary specimens only if they had a previously reported metastasis at the said site. **b** Forest plots from multivariate linear models testing the association between the biopsied site (rows) and TMB scores across different cancer types (columns). Each cancer type and tissue are modeled independently. For each site, patients included in the reference tissues (represented by a dark gray box with a dot at 0) used for modeling had a previously reported metastasis at that site. The dots and color gradient represent the average difference between the tissue and a reference tissue. The horizontal lines represent the 95% confidence interval of the average difference. The transparency of the box colors is adjusted according to the significance (*p* value) of the association. Patient counts for each cancer type and biopsy site are provided in Supplementary Data 5. **c** Paired analysis of brain and lung biopsies in non-small cell lung cancer patients biopsied at both sites within 90 days.

and Supplementary Data 3). For example, in patients diagnosed with pancreatic cancer, generally a low-TMB cancer type, samples from the lymph nodes were 5 times more likely to be assessed as TMB-high versus samples from the pancreas (7.2% [37/513] versus 1.5% [103/6895]. OR 5.14; 95% CI [3.45–7.50]; *p* < 0.01). Our results confirm the impact of the biopsied site on TMB variability and its implication in clinical decision-making based on a fixed cutoff. Moreover, our findings show that the differences are observed across the TMB spectrum.

**TMB differences are not driven by non-metastatic patients.**
Our analysis compared groups of patients based on the biopsy site tested. It is possible that these groups were unbalanced for important variables. For example, all the NSCLC patients with a brain biopsy, by definition, had a brain metastasis. However, only some of the patients with a primary lung biopsy also had brain metastases. To address this variable, we used clinical annotations from the Flatiron-Foundation Medicine CGDB (Flatiron Health, New York, NY). When comparing primary specimens to metastatic specimens, we only retained subjects who had a previously reported lesion at that given metastatic site. In other words, this excluded primary specimens with no documentation of

metastasis at the compared metastatic site (Fig. 3a). The data available for patients diagnosed with breast, NSCLC, and melanoma across a reduced set of metastatic sites yielded robust results using this conservative approach (Fig. 3b and Supplementary Data 4 and 5). When controlling for the presence of a metastasis, NSCLC patients biopsied from a brain or adrenal gland site had higher TMB scores than patients biopsied from their lung (mean difference 2.66 mut/Mb; 95% CI [0.87–4.47]; *p* < 0.01, 6.05 mut/Mb; 95% CI [3.33–8.77]; *p* < 0.01, respectively).

Finally, we identified a set of 9 patients biopsied from both lung and brain sites, <90 days apart. A paired analysis of biopsy sites yielded a consistent result, with brain biopsies having consistently higher TMB scores (*p* = 0.03; Fig. 3c).

## Discussion
Our comprehensive, pan-cancer study comprised of 137,771 specimens from 47 tissues in 12 indications from a real-world cohort highlights the impact of the biopsy site on TMB scores in a cancer-specific manner.

The comparison of metastatic and primary lesions is an active area of research[4]. Our results are consistent with previous findings of higher TMB scores in brain and adrenal gland metastases

and lower TMB scores in bone metastases in NSCLC patients[17]. We expand into pan-cancer profiling of TMB across multiple biopsy sites. We show that comparing metastatic and primary tissues fails to account for important site-specific effects and thus does not tell the whole story. For example, whereas no differences were observed between metastatic and primary specimens of bladder cancer patients (Fig. 2a), lymph node metastases had higher TMB scores and lung metastases had lower TMB scores. Thus, when biopsy sites were considered together, they effectively masked site-specific differences. Generally, metastatic sites had higher TMB scores than their respective primary site. This is exemplified by metastatic specimens from breast, endometrial, and ovarian cancer that all have higher TMB scores than their respective primary site (Fig. 2d). This trend could be the result of the expansion of one or a few metastasis seeding cells, carrying multiple subclonal or private mutations, otherwise not observable when sequencing the bulk primary tumor, thus resulting in higher TMB. In other indications, however, we observed striking differences of TMB scores at specific sites highlighting the cancer type and tissue specificity of TMB differences.

Our study shows that in clinical practice, the locations of biopsied tissues sent for biomarker testing are highly diverse. This diversity varies by indication, likely driven by the metastatic patterns of each cancer type. Moreover, the biopsy site sent for testing may also be driven by clinical practice considerations. These include the ease—or need—to resect certain metastases as part of patient care. Finally, the amount and quality of tissue obtained by different biopsy or resection methods, in addition to methods used to process tissue from different sites, such as hard acid bone decalcification, may also bias the observed distribution of biopsied sites. These considerations and, therefore, the resulting biopsied sites are likely to differ between the real-world setting and clinical trials. Thus, our study provides important context for considering the biopsied site of target lesions when assessing molecular biomarkers.

Given the limited clinical data at hand, our study is not suited to address the potential impact of prior treatment on TMB measurements. Although treatment-induced mutations has been reported in some cases, most notably in Temozolomide-treated glioblastoma patients[23], the impact of treatment exposure on TMB is not clear. Interestingly, Christensen et al. observed that the impact of 5-fluorouracil treatment on TMB differed extensively among patients, with only a subset of patients showing differences in TMB[24]. More recently, a study in 970 NSCLC patients, treated with cytotoxic chemotherapy or radiation therapy, found no difference in TMB in samples from patients biopsied pretreatment compared to those of patients biopsied after treatment[25].

As genomic testing is increasingly becoming part of routine oncological clinical decision-making, factors that influence the observed genomics of a patient's tumor merit serious consideration. Although out of scope for this study, our results warrant a careful analysis of the associations between response to immunomodulatory drugs and TMB scores, controlling for the biopsied site. Importantly, our results have direct implications for the interpretation of TMB scores in clinical practice as decisions are binary and based on a fixed threshold. Prior work highlighted the challenges caused by sampling from a single region of a given tumor by highlighting intratumor mutation variability[26]. Our findings supplement the need for further profiling of the mutational landscape of cancer material, highlighting the site of biopsy as an important factor of variability. A sufficiently powered cohort of biopsies from multiple sites of metastases in the same patients, taken at the same time, would allow one to perform the gold-standard analysis.

## Data availability

Source data for the main figures in the manuscript is available as Supplementary Data 1–5. The data that support the findings of this study are available from Foundation Medicine, but restrictions apply to the availability of these data, which were used under license for the current study, and so are not publicly available. Academic researchers can gain access to Foundation Medicine data in this study by contacting the Foundation Medicine authors and filling out a study review committee form. Any interested readers and their institutions will be required to sign a data transfer agreement.

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

## Author contributions

S.P.-C. and A.M.W. conceived and planned the analyses. S.P.-C., J.F.H. and L.A.A. carried out the analyses. S.H.R. provided clinical interpretation of the results. All authors contributed to the interpretation of the results, provided critical feedback, and participated in manuscript preparation.

## Competing interests

S.P.-C. and A.M.W. are employees of Bristol Myers Squibb. J.F.H., S.H.R. and L.A.A. are employees of Foundation Medicine.
