## [Peer Review File · Communications Medicine]

Reviewers' comments:

Reviewer #1 (Remarks to the Author):

The present manuscript utilizes the large dataset of tumors profiled by Foundation Medicine to investigate and compare site specific TMB differences for each cancer type. The authors conclude that the specific metastatic site might impact the TMB value for some cancer types and that thus, the site of biopsy could impact clinical decision making. This is an interesting aspect, however the different genomic composition of primary vs metastatic sites has previously been shown in the TRACERx studies (eg. <https://www.nature.com/collections/haffgaicaf>) and even intra-tumoral TMB differences have been studied (doi: 10.1016/j.jtho.2019.07.006). However, the authors rely on a very high number of samples thereby certainly adding novelty to the scientific literature.

The manuscript is very well written, and the authors made their best efforts to account for probable confounding variables that would have impacted the analysis and based on the very high amount of samples included in the analysis, the result should be quite robust. Additionally, the Figures are well prepared and understandable.

However, many of the differences observed are very minor (less than 1 mutations TMB difference) and despite being statistically significant one could consequently question the clinical significance. I would argue that the most important result is thus how many patients per site/cancer have a TMB ≥ 10 as this threshold was FDA approved. While the authors shortly report on that measurement, giving an example, I think it is very important to highlight this information for all the tumor types and to attach the data prominently in the manuscript.

Minor:

- There is a small typo in the first paragraph of the fourth page ["mub" instead of "mut"]:
"...from the adrenal gland and brain metastases had strikingly higher TMB scores relative to lung biopsies (mean difference 3.88 mut/Mb; 95%CI [2.77 – 4.99]; $p < 0.01$, 2.77 mub/Mb; 95%CI [2.17 – 3.39]; $p < 0.01$, respectively),..."

Reviewer #2 (Remarks to the Author):

In this paper, the authors assess Tumour Mutation Burden per Foundation One testing in a large real world cohort across cancer types and metastatic sites. They report that some cancers appear to have some differences in TMB between primary and metastases, with the strongest difference in melanoma (-1.55). Other cancer types showed higher TMB score in metastases compared to primary. There was also some cancer type specific variation in TMB between metastatic sites. However, they did not report a systematic difference between organ sites that was agnostic of cancer type.

Major comments

- The authors need to show that the performance of the TMB estimated from FM panel is adequate at the lower end of TMB spectrum – here a small number of mutations that are false negatives or

positives can make a large difference.

- They also need to show how the sequencing coverage and tumour purity might affect their results

- The findings in bone metastases could reflect the lower yield and poorer quality of DNA usually extracted from bone relative to other sites- this will ultimately decrease the sensitivity for variant detection.

- Other important factors may significantly increase mutation burden, such as prior chemotherapy and radiotherapy. Have the authors considered this or do they have access to whether the samples have been exposed to prior treatment, in order to potentially adjust for this?

The differences reported here between TMB between primary and mets can, at least partly, be explained by prior exposure to anticancer therapy. In cancers such as breast and NSCLC, many pts have adjuvant treatment which will increase the mutation burden. In these cancers, they report TMB higher in mets than in primary – which would be consistent with this. In contrast, in Small cell lung cancer (where most pts present with metastatic disease) and melanoma, biopsy of metastatic disease often occurs in the treatment naïve setting. In these cancer types, TMB in mets was lower than TMB in primary.

Could the authors discuss this further?

- The authors state that in general, metastatic sites have a higher TMB than the primary – could the authors hypothesise as to any other reasons why?

Minor comments

The authors state that “sufficiently powered cohort of biopsies from multiple sites of the same patients, taken at the same time, would allow one to perform the gold-standard analysis. “. This was actually addressed in Litchfield et al Cell Reports 2020– please cite

In the methods, the authors mention that there were patients with multiple specimens, but only kept the most recent specimen. It would be interesting to include these to assess intrapatient heterogeneity between tumour sites, especially if these were performed at the same time point.

We note that primary brain malignancies are not represented in this cohort – is there any reason for this?

Some sentences are difficult to follow, in particular please review page 4, paragraph 3, and figure 3 – e.g. ‘NSCLC patients biopsied from a brain or adrenal gland site had higher TMB scores than patients biopsied from their lung that had a metastasis reported at those sites’

Introduction - Please reference study for approval of TMB high as a biomarker

Figure 2a – legend should read ‘metastases’, not ‘metastatic’

Figure 2b – can the figure be weighted to reflect the proportion of samples from each site?

Reviewers' comments:

Reviewer #1 (Remarks to the Author):

The present manuscript utilizes the large dataset of tumors profiled by Foundation Medicine to investigate and compare site specific TMB differences for each cancer type. The authors conclude that the specific metastatic site might impact the TMB value for some cancer types and that thus, the site of biopsy could impact clinical decision making. This is an interesting aspect, however the different genomic composition of primary vs metastatic sites has previously been shown in the TRACERx studies

(eg. <https://www.nature.com/collections/haffgaicaf> [[nam02.safelinks.protection.outlook.com](https://doi.org/10.1016/j.jtho.2019.07.006)] [[nam02.safelinks.protection.outlook.com](https://doi.org/10.1016/j.jtho.2019.07.006)]) and even intra-tumoral TMB differences have been studied (doi: 10.1016/j.jtho.2019.07.006). However, the authors rely on a very high number of samples thereby certainly adding novelty to the scientific literature.

The manuscript is very well written, and the authors made their best efforts to account for probable confounding variables that would have impacted the analysis and based on the very high amount of samples included in the analysis, the result should be quite robust. Additionally, the Figures are well prepared and understandable.

We thank the reviewer for their positive comments on the interesting nature, novelty, and overall clarity of our manuscript. As suggested by the reviewer, we have added text and references pointing to the prior work on intra-lesion heterogeneity by the TRACERx consortium, in the last paragraph of the introduction.

However, many of the differences observed are very minor (less than 1 mutations TMB difference) and despite being statistically significant one could consequently question the clinical significance. I would argue that the most important result is thus how many patients per site/cancer have a TMB ≥ 10 as this threshold was FDA approved. While the authors shortly report on that measurement, giving an example, I think it is very important to highlight this information for all the tumor types and to attach the data prominently in the manuscript.

We thank the reviewer for this suggestion. We agree that this is an important clinical implication of our results and as such have performed the suggested analysis, comparing the percentage of specimens assessed as TMB-High between biopsied sites, across cancer types. The results are presented in Figure 2 in a new panel E, with accompanying Supplementary Table 3 Accordingly, we added new sections in the Results and Methods portions of the manuscript.

Minor:

- There is a small typo in the first paragraph of the fourth page [“mub” instead of “mut”]:
“...from the adrenal gland and brain metastases had strikingly higher TMB scores relative to lung biopsies (mean difference 3.88 mut/Mb; 95%CI [2.77 – 4.99]; $p < 0.01$, 2.77 mub/Mb; 95%CI [2.17 – 3.39]; $p < 0.01$, respectively),...”

We thank the reviewer for catching this typo and have corrected it

Reviewer #2 (Remarks to the Author):

In this paper, the authors assess Tumour Mutation Burden per Foundation One testing in a large real world cohort across cancer types and metastatic sites. They report that some cancers appear to have some differences in TMB between primary and metastases, with the strongest difference in melanoma (-1.55). Other cancer types showed higher TMB score in metastases compared to primary. There was also some cancer type specific variation in TMB between metastatic sites. However, they did not report a systematic difference between organ sites that was agnostic of cancer type.

Major comments

- The authors need to show that the performance of the TMB estimated from FM panel is adequate at the lower end of TMB spectrum – here a small number of mutations that are false negatives or positives can make a large difference.

We thank the review for this important comment. In response to this comment and related ones, we have performed an additional set of analyses assessing the difference in percent of patients being assessed as TMB-High (≥ 10 mut/Mb) across biopsied sites. This approach eliminates the effect of noise in lower end of the TMB spectrum samples. Those new results, presented in Figure 2 in a new panel E and Supplementary Table 3 robustly replicate previous findings using TMB as a continuous variable.

In the technical specifications, a TMB cutoff of 10 has ~90% positive percent and negative percent agreement with whole exome sequencing (note the technical specifications have been reviewed by the FDA and can be found here (Table 7 pg 17): https://info.foundationmedicine.com/hubfs/FMI%20Labels/FoundationOne_CDx_Label_Technical_Info.pdf)

- They also need to show how the sequencing coverage and tumour purity might affect their results

The reviewer raises a potential problem most prominently highlighted in Anagnostou et al Nature Cancer 2020. Tumor purity is more consequential for TMB measurements in whole exome sequencing because coverage is generally lower and less even than panel sequencing. The in silico analysis in Anagnostou used 200X with an allele frequency cutoff of 10%. Median coverage in this study coverage exceeds 500X and the allele frequency cutoff is 5%. In our precision analysis of TMB (Tech specs Table 13-3 pg25), reproducibility was >99% with half the samples having a near limit of detection tumor purity. According to Table 17-2 pg30, in the worst case scenario the limit of detection for TMB >10 is 28% tumor purity.

Finally, to address this comment, we have redone all analyses in this paper excluding samples below 30% purity. Those changes are reflected throughout the manuscript (Redone Figures 1-3

and Supplementary Tables 1-5). This filtering did not meaningfully change any results previously presented. We have also clarified the Methods section.

- The findings in bone metastases could reflect the lower yield and poorer quality of DNA usually extracted from bone relative to other sites- this will ultimately decrease the sensitivity for variant detection.

Bone metastases can certainly be treated more harshly than other biopsy sites, although prostate is another difficult biopsy site. Every sample is hand reviewed by a pathologist to make sure it is suitable for sequencing. Poor quality DNA also triggers a number of QC flags which we used as an exclusion criteria for this study. One such flag is to set TMB to 'None', if short variant artifacts are detected. We have clarified those points in the Methods section.

- Other important factors may significantly increase mutation burden, such as prior chemotherapy and radiotherapy. Have the authors considered this or do they have access to whether the samples have been exposed to prior treatment, in order to potentially adjust for this?

The differences reported here between TMB between primary and mets can, at least partly, be explained by prior exposure to anticancer therapy. In cancers such as breast and NSCLC, many pts have adjuvant treatment which will increase the mutation burden. In these cancers, they report TMB higher in mets than in primary – which would be consistent with this. In contrast, in Small cell lung cancer (where most pts present with metastatic disease) and melanoma, biopsy of metastatic disease often occurs in the treatment naïve setting. In these cancer types, TMB in mets was lower than TMB in primary.

Could the authors discuss this further?

The reviewer raises an important point regarding the potential causes of the observed TMB differences. Unfortunately, given the nature of the real-world data at hand, we do not have comprehensive treatment data for those patients. Moreover, prior treatment could explain global TMB differences, but not the much stronger, site-specific effects we observed. We have added to the Discussion section of the manuscript to address the putative impact of treatment on our results and acknowledge that further research into the mechanisms behind the differences we observed is of high scientific interest.

- The authors state that in general, metastatic sites have a higher TMB than the primary – could the authors hypothesise as to any other reasons why?

Although we can only make conjectures at this point, it is possible that the metastatic seeding by one or a few cells carrying subclonal mutations (not otherwise observable when sequencing the bulk primary tumor). We have a statement to address this in the Discussion section of the manuscript

Minor comments

The authors state that “sufficiently powered cohort of biopsies from multiple sites of the same patients, taken at the same time, would allow one to perform the gold-standard analysis. “. This was actually addressed in Litchfield et al Cell Reports 2020– please cite

We thank the reviewer for identifying this important resource. We have reworded the last paragraph of the Discussion to highlight their findings.

In the methods, the authors mention that there were patients with multiple specimens, but only kept the most recent specimen. It would be interesting to include these to assess inpatient heterogeneity between tumour sites, especially if these were performed at the same time point.

As part of standard of care, which is the source of the specimens profiled in this study, patients undergoing repeat genetic testing is excitingly rare. For example, out of the 57,796 NSCLC patients in our dataset, only 9 patients had both lung and brain specimens (the highest tissue-pair). We presented this paired analysis, which supports our previous in Figure 3C.

We note that primary brain malignancies are not represented in this cohort – is there any reason for this?

Brain malignancies do not tend to metastasize and thus were excluded from the dataset

Some sentences are difficult to follow, in particular please review page 4, paragraph 3, and figure 3 – e.g. ‘NSCLC patients biopsied from a brain or adrenal gland site had higher TMB scores than patients biopsied from their lung that had a metastasis reported at those sites’

We agree with the reviewer and have clarified this statement

Introduction - Please reference study for approval of TMB high as a biomarker

We thank the reviewer for identifying this omission and have added the relevant reference.

Figure 2a – legend should read ‘metastases’, not ‘metastatic’

We thank the reviewer for catching this typo and have corrected it

Figure 2b – can the figure be weighted to reflect the proportion of samples from each site?

We thank the reviewer for the suggestion. In order to provide this important data whilst avoiding burdening the visual for Figure 2B/C, we have added Supplementary Table 3 which details how many patients are assessed as TMB-High across all tissues, for all indications.

REVIEWERS' COMMENTS:

Reviewer #1 (Remarks to the Author):

The authors have addressed my comments well and there are no further requests from my side. Consequently, I do recommend this manuscript for publication and congratulate the authors on their work.

Reviewer #2 (Remarks to the Author):

Thank you for the opportunity to re-review this submission. We thank the authors for addressing the points raised. Regarding performance at the lower end of TMB, in addition to analysing it as a continuous variable, the authors have added analysis of TMB with cut-off of 10 (Fig 2E) as a binary approach, that shows the additional % of samples classified as TMB-high compared to the primary. They have addressed our query regarding tumour purity by repeating analyses with only samples >30% tumour purity.

We would like to raise a couple of other minor points below:

- Discussion P6

The authors reference Jonna et al 2021 to state that no difference in TMB was found between pre and post treatment samples. However, the samples used in this study were not paired pre and post treatment samples, nor from the same metastatic site from the same patient – thus we would have some reservations in drawing this conclusion.

-P4 Subheading 'TMB differences are not driven by non-mestatic patients' – spelling error metastatic.

REVIEWERS' COMMENTS:

Reviewer #1 (Remarks to the Author):

The authors have addressed my comments well and there are no further requests from my side. Consequently, I do recommend this manuscript for publication and congratulate the authors on their work.

We thank the reviewer for their comments which benefited the manuscript and endorsement for publication.

Reviewer #2 (Remarks to the Author):

Thank you for the opportunity to re-review this submission. We thank the authors for addressing the points raised. Regarding performance at the lower end of TMB, in addition to analysing it as a continuous variable, the authors have added analysis of TMB with cut-off of 10 (Fig 2E) as a binary approach, that shows the additional % of samples classified as TMB-high compared to the primary. They have addressed our query regarding tumour purity by repeating analyses with only samples >30% tumour purity.

We thank the reviewer for their comments which benefited the manuscript and endorsement for publication.

We would like to raise a couple of other minor points below:

- Discussion P6

The authors reference Jonna et al 2021 to state that no difference in TMB was found between pre and post treatment samples. However, the samples used in this study were not paired pre and post treatment samples, nor from the same metastatic site from the same patient – thus we would have some reservations in drawing this conclusion.

We have edited the last sentence of 4th paragraph in Discussion to avoid confusion about whether samples are taken from the same patients.

-P4 Subheading ‘TMB differences are not driven by non-mestatic patients’ – spelling error metastatic.

Thank you for spotting this typo, we have corrected it.